Spatial and temporal distribution of the invasive lionfish Pterois volitans in coral reefs of Tayrona National Natural Park, Colombian Caribbean

Bayraktarov Elisa 1 4 elisa.bayraktarov@uni-bremen.de
Alarcón-Moscoso Javier 2
Polanco F. Andrea 2
Wild Christian 1 3
1 Coral Reef Ecology Group (CORE), Leibniz Center for Tropical Marine Ecology , Bremen , Germany
2 Instituto de Investigaciones Marinas y Costeras ‘José Benito Vives de Andréis’ (Invemar) , Santa Marta , Colombia
3 Faculty of Biology and Chemistry (FB2), University of Bremen , Bremen , Germany
Toonen Robert
4 Current affiliation: Global Change Institute, The University of Queensland, Brisbane, Australia

Electronic publication date: 2014 May 22
Publication date: 2014
Volume: 2
Electronic Location ID: e397
Received 2014 Mar 3; Accepted 2014 May 7
Copyright: © 2014 Bayraktarov et al.
Copyright year: 2014
Copyright holder: Bayraktarov et al.
License: This is an open access article distributed under the terms of the Creative Commons Attribution License, which permits unrestricted use, distribution, reproduction and adaptation in any medium and for any purpose provided that it is properly attributed. For attribution, the original author(s), title, publication source (PeerJ) and either DOI or URL of the article must be cited.
License URL: https://creativecommons.org/licenses/by/4.0/

Keywords: Invasive lionfish, Colombian Caribbean, Tayrona National Natural Park, Spatial and temporal distribution, Removals effects, Body lengths

Funding: German Academic Research Service (DAAD) through the German–Colombian Center of Excellence in Marine Sciences (CEMarin) This study was supported by the German Academic Research Service (DAAD) through the German–Colombian Center of Excellence in Marine Sciences (CEMarin) under the coordination of Thomas Wilke. The funders had no role in study design, data collection and analysis, decision to publish, or preparation of the manuscript.

==============================
The lionfish Pterois volitans is an invasive species throughout the Western Atlantic that disturbs functioning of local ecosystems such as coral reefs via fast and intense consumption of small fish and invertebrates. In 2009, lionfish populated the bays of Tayrona National Natural Park (TNNP), a biodiversity hotspot in the Colombian Caribbean that is strongly influenced by changing environmental conditions due to a rainy and dry season. So far, the spatial and temporal distribution of P. volitans in the bays of TNNP is unknown. Therefore, this study assessed the abundance and body lengths of P. volitans during monthly surveys throughout the year 2012 in four bays (thereof two bays where lionfish removals were undertaken) of TNNP at 10 m water depth in coral reefs using transect tools. Findings revealed lionfish abundances of 2.9 ± 0.9 individuals ha−1 with lengths of 20–25 cm for TNNP, hinting to an established, mostly adult local population. Actual TNNP lionfish abundances are thereby very similar to those at Indo–Pacific reef locations where the invasive lionfish formerly originated from. Significant spatial differences for lionfish abundances and body lengths between different bays in TNNP suggest habitat preferences of P. volitans depending on age. Lionfish abundances were highly variable over time, but without significant differences between seasons. Removals could not reduce lionfish abundances significantly during the period of study. This study therefore recommends improved management actions in order to control the already established invasive lionfish population in TNNP.

Introduction

The Indo–Pacific lionfish Pterois volitans belongs to the family Scorpaenidae and is an invasive marine fish that was introduced in the Western Atlantic during the 1980’s (Whitfield et al., 2002; Morris & Whitfield, 2009; Schofield, 2010; Albins & Hixon, 2011; Arias-González et al., 2011). Lionfish in invaded areas have many advantages over native fauna e.g., their generalist diet on a variety of smaller fishes, shrimps and small mobile invertebrates (Morris & Akins, 2009), defensive venomous spines (Morris & Whitfield, 2009; Albins & Hixon, 2011), rapid growth (Albins & Hixon, 2011), low parasite load (Morris, 2009), and habitat generality (Barbour et al., 2010; Albins & Hixon, 2011). These characteristics make lionfish dramatically decrease local populations in invaded areas (Albins & Hixon, 2008; Arias-González et al., 2011) with strong implications for the trophic web structures of the marine ecosystem (Mack et al., 2000). Additional features of lionfish such as high fecundity (Morris, 2009; Morris & Whitfield, 2009), effective larval dispersal mechanisms (Morris & Whitfield, 2009), and efficient predation (Albins & Hixon, 2008; Albins & Hixon, 2011; Côté & Maljković, 2010) increase their probability of invasion success.

In Colombia, lionfish arrived to the oceanic islands of San Andrés and Providencia in 2008 and invaded the entire continental coast of the country in the course of the following year. For the Tayrona National Natural Park (TNNP; Fig. 1) on the northeast Colombian coast, the presence of P. volitans was first recorded in May–July 2009 at water depths between 12 and 20 m over coral patches (González et al., 2009). In 2010, juvenile P. volitans (3–10 cm lengths) were observed in the mangrove ecosystem of Chengue Bay (Arbeláez & Acero P, 2011). The ecological consequences of lionfish are of particular interest for the TNNP area due to its major coastal biodiversity (Garzón-Ferreira & Cano, 1991). The TNNP is a fishing restricted area administrated by the National Natural Parks of Colombia dealing with all territories of marine parks and reserves. Generally, the TNNP includes different coastal bays with complex structural bottoms offering heterogeneity of habitats suitable for a high marine biodiversity. A record diversity was reported especially for macroalgae (Bula-Meyer & Norris, 2001; Diaz-Pulido & Garzón-Ferreira, 2002; Diaz Pulido & Díaz Ruiz, 2003) but also for other marine organisms (e.g., mollusks; Díaz, 1995; Diaz-Pulido, 1998).

Figure 1 Area of survey in the Tayrona National Natural Park (TNNP).

The points indicate the sampling locations at the western and eastern flank of each bay. Source: Laboratorio de Sistemas de Información LabSIS, INVEMAR, 2013.

So far, little is known about the spatial and temporal distribution of lionfish in TNNP. Therefore, the aim of the present study was to assess monthly P. volitans abundances and estimated body lengths throughout one year (2012) for four bays within TNNP. The first objective was to compare lionfish data from TNNP with other invaded areas and also with its native locations in the Indo–Pacific. The second objective was to determine whether lionfish abundances change over time and if differences between a rainy and dry season, coinciding with a seasonal upwelling, exist. This objective focused on possible effects of strong seasonal variation in physicochemical parameters (temperature, salinity, wind, water currents, surplus of inorganic nutrients; Bayraktarov et al., 2013; Bayraktarov, Pizarro & Wild, 2014) during seasonal upwelling on lionfish distribution, as concluded for the factor temperature by the experimental study of Kimball et al. (2004) indicating ceased feeding of lionfish at 16 °C with lethal consequences at 10 °C. The third objective addressed the efficiency of management actions (removals) that started in May 2012 in two TNNP bays by comparing lionfish abundances before removals with data collected after removals for two of the four bays. The present study provides recent and comprehensive lionfish distribution data and establishes an actual baseline with high temporal and spatial resolution for TNNP reefs in the Colombian Caribbean. Further needs of management actions to control the already established invasive lionfish population in TNNP are discussed.

Materials and Methods

Study site

All necessary permits were obtained for the described study by Instituto de Investigaciones Marinas y Costeras ‘José Benito Vives de Andréis’ (Invemar) in Santa Marta, Colombia which complied with all relevant regulations (decree # 302 and # 309).

The Tayrona National Natural Park (TNNP) is located on the northeastern coast of Colombia, between 11°17′–11°22′N and 73°53′–74°12′W (Fig. 1). The region contains a rocky coastline with capes, inlets and bays with sandy beaches covering over 40 km (Garzón-Ferreira & Cano, 1991; Díaz et al., 2000; Martínez & Acosta, 2005). The area of survey included the main TNNP bays Chengue, Gayraca, Neguanje, and Cinto (Fig. 1) which experience strong seasonal changes due to a rainy season (>80% of the annual rainfall, May–November) and a dry season (December–April) characterized by a seasonal upwelling with strong changes in temperature (decrease from 28 to 21 °C), salinity (increase from 33 to 38), increased wind and water currents (Salzwedel & Müller, 1983; Mesa, Poveda & Carvajal, 1997; Bayraktarov et al., 2013; Bayraktarov, Pizarro & Wild, 2014). Increased concentration of inorganic nutrients and chlorophyll a during periods of upwelling (dry season) result in mesotrophic conditions, compared with oligotrophic settings during the non-upwelling periods (rainy season; Franco-Herrera et al., 2007; Arévalo-Martínez & Franco-Herrera, 2008; Bayraktarov, Pizarro & Wild, 2014).

Coral reef formations can be found growing on both sides of each bay between water depths of 5 to 20 m (Werding & Erhardt, 1976; Werding & Sánchez, 1989; Garzón-Ferreira & Cano, 1991) and represent a habitat for over 180 reef fish species (Olaya-Restrepo, Reyes-Nivia & Rodríguez-Ramírez, 2008). Additionally, the bays harbor mangrove ecosystems and seagrass beds (Fig. 1; Garzón-Ferreira & Cano, 1991).

Lionfish assessment in space and time

In order to address the goals of the study, P. volitans abundances were monitored monthly in four bays of TNNP. Surveys comprised monitoring along line transects of 50 m length and 5 m width in triplicates that were located at the western and eastern flank of each bay (Fig. 1) in order to encompass a representative area for lionfish distribution. Transects were located at water depths between 9 and 11 m, parallel to the coastline, and were separated by >5 m to ensure independence between samples. The investigated area covered 1500 m2 per bay and a total of 6000 m2 within the TNNP. The method of visual census was applied by SCUBA (English, Wilkinson & Baker, 1997; Lang et al., 2010). The total number of P. volitans observed during a time of 25 min per replicate was counted (Morris, 2009). Places where lionfish may hide such as holes and cavities between rocks and coral framework were carefully examined. Estimated total body lengths (TL) of lionfish were recorded in situ in 5 cm intervals for each localized individual from the tip of the snout to the tip of the caudal fin. The surveys were performed between the second and the third week of each month, between 8:00 am and 3:00 pm.

We were informed that lionfish removals were planned to start in May 2012 as a joint project between Universidad Nacional de Colombia, Universidad Jorge Tadeo Lozano, Universidad del Magdalena, Instituto de Investigaciones Marinas y Costeras ‘José Benito Vives de Andréis’ (Invemar) and the National Natural Parks of Colombia. Removals were performed monthly by spearing and netting at variable depths (5–25 m) by SCUBA diving, and to our knowledge, exclusively in the TNNP bays Chengue and Cinto. Additional unregistered removals of lionfish by dive centers or fishermen could not be considered in the present survey.

Data analysis

Mean monthly abundances of P. volitans in the TNNP bays Chengue, Gayraca, Neguanje and Cinto (Fig. 2A) were calculated from data collected over 12 months with a replication of 6 transects per bay and month and were converted into individuals per hectare (ind ha−1; Fig. 2A). Monthly abundance before onset of removals in May were estimated by calculating the abundance for the time period January to April, whereas lionfish data collected between May and December were used to determine the monthly mean abundance after removals (Fig. 2B). For calculation of the temporal lionfish distribution, all lionfish transect data were aggregated per month resulting in a replication of n = 24 transects (Fig. 3A). Annual mean abundance was determined for each bay and the whole TNNP area by pooling the data collected over 12 months resulting in a transect replication of n = 69 transects for Chengue and Neguanje, and n = 72 for Gayraca and Cinto (total of 282 transects). Mean estimated sizes of lionfish (Fig. 3B) were calculated from total estimated body lengths of fishes observed along the transects in the respective bay.

Differences in P. volitans abundances between bays and months were tested by a Generalized Linear Model (GLM) for Poisson-distributed data and the software R (R Development Core Team, 2008). Multiple comparisons between bays (Chengue, Gayraca, Neguanje and Cinto) and months were performed by a Tukey’s Honestly Significant Difference (HSD) post hoc test on data before (January–April; 93 transects) and after onset of removal (May–December; 189 transects). For a quantification of possible removal effects, GLMs for Poisson-distributed data and Tukey’s HSD post hoc tests were performed before and after removal in the bays Chengue and Cinto, individually.

Figure 2 Lionfish abundances in Tayrona National Natural Park before and after removal.

(A) Abundances (monthly mean ± SE) of Pterois volitans in the bays Chengue, Gayraca, Neguanje and Cinto throughout the months of 2012. The red line indicates the starting period of monthly removals (May 2012) from the bays Chengue and Cinto. Removal bays (Chengue and Cinto) are indicated by solid symbols, while non-removal bays have open symbols. (B) Mean lionfish abundances (+SE) before (January–April) and after removal (May–December). Abbreviations: Chengue, (Ch); Gayraca, (Ga); Neguanje, (Ne); and Cinto, (Ci).

Figure 3 Monthly abundances of Pterois volitans.

(A) Monthly mean + SE; aggregate of four bays for Tayrona National Natural Park and (B) estimated body lengths for the bays Chengue, Gayraca, Neguanje, and Cinto. Missing error bars represent sample sizes which did not allow the calculation of a mean and SE at some locations and months.

Results

Spatial distribution of lionfish in TNNP

Throughout the year 2012, 123 individuals of Pterois volitans were counted during 12 months in four bays over a total monitored area of 6000 m2. Before removals, lowest mean lionfish abundance was found in Chengue Bay with 1.7 ± 1.0 ind ha−1 (monthly mean ± SE for the months January–April), followed by Cinto with 2.5 ± 0.3, and Neguanje with 3.9 ± 1.0 ind ha−1. Highest numbers of monthly lionfish were present in Gayraca with 5.8 ± 3.6 ind ha−1. Significant differences in lionfish abundances during the months before removal were present between the bays Chengue and Gayraca (GLM, Tukey’s HSD post hoc, p = 0.033) with higher lionfish abundance in Gayraca. After onset of monthly removal in May, lowest monthly lionfish abundance was observed in Neguanje with 0.8 ± 0.5 ind ha−1, followed by Chengue with 1.3 ± 0.6 and Cinto with 3.9 ± 2.0 ind ha−1. Highest monthly lionfish abundance was still observed in Gayraca Bay with 4.4 ± 1.7 ind ha−1. After removal, significant differences were found between Chengue and Cinto (p = 0.017), Chengue and Gayraca (p = 0.004), Gayraca and Neguanje (p < 0.001) and Neguanje and Cinto (p = 0.003).

Our monthly lionfish censuses demonstrated temporal and spatial variability in lionfish abundances among TNNP bays, which varied between 0 and 16.7 ind ha−1 (Fig. 2A). In Chengue (a removal bay), lionfish abundances were below 5 ind ha−1 (monthly mean ± SE) until July and disappeared thereafter completely until December, where 1.1 ± 1.1 ind ha−1 were registered. Highest abundances were observed in Gayraca during January with 16.7 ± 10.3 ind ha−1 and August with 12.2 ± 4.6 ind ha−1, while intermediate abundances were present during September with 10.0 ± 5.6 ind ha−1 and December with 7.8 ± 6.5 ind ha−1 within this bay. In Neguanje, highest lionfish abundances were recorded in January with 5.6 ± 5.6 ind ha−1 and February with 5.6 ± 3.6 ind ha−1. Here, no lionfish were observed between July and December, except for September when 2.2 ± 2.2 ind ha−1 were registered. In Cinto, lionfish abundances peaked during September with 16.7 ± 5.6 ind ha−1 and June with 6.7 ± 2.4 ind ha−1, but were otherwise below 5 ind ha−1. Mean lionfish abundance before removal was 10.0 ± 5.8 ind ha−1 for Chengue, 35.0 ± 21.8 ind ha−1 for Gayraca, 21.7 ± 6.9 ind ha−1 for Neguanje, and 15.0 ± 1.7 ind ha−1 for Cinto (Fig. 2B). Lionfish abundance for the months during which removal actions were performed changed the values to 5.0 ± 3.3, 28.3 ± 9.8, 5.0 ± 2.7, and 23.3 ± 12.1 ind ha−1, respectively (Fig. 2B).

Temporal distribution of lionfish in TNNP

On the temporal scale, highest abundance of lionfish was observed in September with 7.2 ± 2.4 ind ha−1 (monthly TNNP mean ± SE; Fig. 3A) and January with 6.4 ± 3.0 ind ha−1; lowest during November with 0.3 ± 0.3 ind ha−1 and July with 0.8 ± 0.6 ind ha−1. Significant differences between months were present between September and April (GLM, Tukey’s HSD post hoc, p = 0.05), July and January (p = 0.04), September and July (p = 0.02), September and May (p = 0.03), and between September and October (p = 0.03). However, lionfish abundances were not significantly different between rainy (May–November) and dry season (December–April).

Largest estimated lionfish body lengths of 40 cm were registered for Cinto in January and August, and Neguanje in September (Fig. 3B). Largest body lengths were present in Gayraca and Cinto with mean sizes of 20–25 cm, followed by Neguanje with 15–20 cm, and smallest in Chengue with 10–15 cm. A total of 75% of all lionfish observed had a body length larger than 17.5 cm TL (20–25 cm) representing the size of 50% maturity for females (Morris, 2009). Before removal, mature lionfish accounted for 80%, and 72% after initiation of removal efforts. Adults were distributed as 13% in Chengue, 51% in Gayraca, 21% in Neguanje, and 15% in Cinto before removal which changed to 4%, 49%, 8%, and 40% after removal, respectively.

The effect of fish removal

Individual GLM analyses showed no significant differences in lionfish abundance before (January–April) and after removal (May–December) for both removal bays, Chengue (GLM, Tukey’s HSD post hoc, p = 0.53) and Cinto (p = 0.25). Since no significant differences were observed before and after removal, transect data were pooled to calculate an annual mean of 2.9 ± 0.9 ind ha−1 (annual mean ± SE) for the TNNP region. The annual mean for Chengue was 1.4 ± 1.3 ind ha−1, 4.9 ± 1.3 ind ha−1 for Gayraca, 1.8 ± 0.6 ind ha−1 for Neguanje, and 3.4 ± 0.8 ind ha−1 for Cinto, respectively (Table 1).

Table 1 Comparison of Pterois volitans abundance in Tayrona National Natural Park (TNNP) with worldwide reports on invaded and native habitats.

Region and year	Habitat for lionfish	Reported abundance
(ind ha−1)	Source	
Chengue Bay (TNNP, Colombian Caribbean), 2012	invasive	1.4 ± 0.6	this study	
Gayraca Bay (TNNP, Colombian Caribbean), 2012	invasive	4.9 ± 1.3	this study	
Neguanje Bay (TNNP, Colombian Caribbean), 2012	invasive	1.8 ± 0.6	this study	
Cinto Bay (TNNP, Colombian Caribbean), 2012	invasive	3.4 ± 0.8	this study	
TNNP, Colombian Caribbean, 2012	invasive	2.9 ± 0.9	this study	
New Providence, Bahamas, Western Atlantic, 2008	invasive	393.3 ± 144.4	Green & Côté (2009)	
Coast off North Carolina, USA, Western Atlantic, 2004	invasive	21.2 ± 5.1	Whitfield et al. (2007)	
Coast off North Carolina, USA, Western Atlantic, 2008	invasive	150	Morris & Whitfield (2009)	
Palau Archipelago, Western Pacific, 2008	native	2.2	Grubich, Westneat & McCord (2009)	
Gulf of Aqaba, Red Sea, 1997	native	∼80	Fishelson (1997)	

Discussion

Spatial and temporal distribution of P. volitans

Our data on P. volitans distribution in Tayrona National Natural Park (TNNP; Colombian Caribbean) show that a local population with mean body length of 20–25 cm has developed in the bays Chengue, Gayraca, Neguanje and Cinto. These lionfish total body lengths (TL) hint to a population mostly dominated by adult fishes that are able to sexually reproduce, based on Morris (2009) who reported 17.5 cm TL as the size of 50% maturity for females.

With an annual mean of 2.9 ind ha−1, lionfish abundance in TNNP was similar to some locations in the Indo–Pacific where it originated from, e.g., Palau Archipelago with 2.2 ind ha−1 (Grubich, Westneat & McCord, 2009), but below ∼80 ind ha−1 reported for the Gulf of Aqaba/Red Sea (Fishelson, 1997). Table 1 shows a comparison of lionfish abundance in TNNP to other invaded and native habitats worldwide, however data should be considered as estimates as methods of monitoring were not always comparable (e.g., rotenone-sampling over small areas; Fishelson, 1997). Lionfish abundances in TNNP were below the values reported for other invaded areas of the Western Atlantic such as the Bahamas with 393 (Green & Côté, 2009) or the coast of North Carolina/USA with 150 ind ha−1 (Morris & Whitfield, 2009) which may be due to the relatively recent invasion of TNNP in 2009 (González et al., 2009) vs. an invasion of the Bahamas in 2004 (Schofield, 2009). High abundances of lionfish in invaded areas are likely the result of unrestricted growth and reproduction due to the availability of food sources and lack of natural predators. Some predators obviously learned to target lionfish as potential prey (Bernadsky & Goulet, 1991; Maljković, Leeuwen & Cove, 2008). So far, two Caribbean large-bodied grouper species, Epinephelus striatus and Mycteroperca tigris, were captured with lionfish in their stomach contents (Maljković, Leeuwen & Cove, 2008). However, E. striatus is one of the species categorized as endangered in the Colombian Caribbean red list of marine fishes (Mejía & Acero, 2002). Mumby, Harborne & Brumbaugh (2011) presented data on the reduction of lionfish biomass by groupers which may thus serve as natural biocontrol of growing lionfish populations. However, the lack of these natural lionfish predators in TNNP (Olaya-Restrepo, Reyes-Nivia & Rodríguez-Ramírez, 2008) and the wider Caribbean Sadovy (2005) is alarming. In contrast to Mumby, Harborne & Brumbaugh (2011), the study of Hackerott et al. (2013) concluded that the abundance of lionfish was not influenced by interaction with native predators in 71 reefs and different biogeographic regions in the Caribbean. The hypothesis of groupers as natural biocontrol against invasive lionfish is currently a subject of active debate (Bruno et al., 2013; Green et al., in press; Mumby et al., 2013). These conflicting results once more stress the necessity of immediate and improved management actions to control further lionfish reproduction and invasion.

Our monthly P. volitans distribution data over four bays in TNNP showed no seasonal pattern between a rainy and a dry season, characterized by seasonal upwelling. The consequently altered environmental conditions (temperature, salinity, water currents, and surplus of inorganic nutrients; Salzwedel & Müller, 1983; Bayraktarov et al., 2013; Bayraktarov, Pizarro & Wild, 2014) did not appear to affect the abundance of lionfish in TNNP. An effect of seawater temperature decrease from 28 to 21 °C (Bayraktarov, Pizarro & Wild, 2014) on lionfish distribution within the area could not be detected. This finding is supported by the laboratory study by Kimball et al. (2004) showing that the critical temperature at which lionfish ceases feeding was 16 °C with lethal consequences at 10 °C, which is more than 10 °C lower that the coldest temperature so far reported for the TNNP upwelling region (20 °C, Bayraktarov, Pizarro & Wild, 2014).

Effect of lionfish removal

Abundances and body lengths of P. volitans for Chengue Bay, in which removals were performed, were smaller than for the uncontrolled Gayraca and Neguanje. However, removals could not effectively reduce lionfish abundances in Cinto which were lower than abundances in Gayraca but higher than in Neguanje. Body lengths found in Cinto corresponded to those in Gayraca. The smallest body lengths observed in Chengue indicate that a mostly juvenile population may have developed in this bay and thereby may indicate a habitat preference dependent on age. However, it cannot be excluded that the smaller body lengths of lionfish in Chengue are a consequence of removal during management actions targeting predominantly larger adult individuals which are easier to observe and catch. Smaller juveniles may hide between the roots of mangroves (Arbeláez & Acero P, 2011) or in crevices and holes of the reef framework which are especially extensive for Chengue Bay (Bayraktarov E, pers. obs., 2010). Additionally, Chengue Bay comprises a highly developed mangrove ecosystem (Garzón-Ferreira & Cano, 1991) which may serve as nursery for lionfish larvae and juveniles. This is further supported by the study of Arbeláez & Acero P (2011), who found lionfish juveniles of 3–10 cm lengths at the submerged roots of the mangroves bordering the entrance to the southern lagoon in Chengue Bay.

Factors affecting fish populations that cannot be excluded are the differences in coral reef complexity between the bays and the potential food sources for lionfish. These important points need to be addressed in further studies.

Our study suggests that management actions for the TNNP require further improvement in terms of removal frequencies and a larger removal area in order to significantly reduce the established lionfish population. Targeted removals were shown to represent a viable strategy in reducing the direct impacts of invasive lionfish on marine ecosystems (Frazer et al., 2012). Frazer et al. (2012) further suggest that management actions should involve long-term monitoring of lionfish distribution, data on recruitment, growth, and reproduction as well as studies on the direct and indirect effects by invasive lionfish on other fish assemblages. The implementation and improvement of management actions in order to preserve the condition of TNNP coral reef ecosystems during P. volitans invasion are crucially essential. The national plan to control and manage lionfish invasion in Colombia is focused on three focal strategy points: (1) realization of fundamental research, (2) implementation of management actions and (3) focus on education and control (MADS et al., 2013). Whereas the first two points are addressed by research groups of universities and institutes, the third point is coordinated by the National Natural Parks of Colombia dealing with all territories of marine parks and reserves. The removal of lionfish outside the marine parks territories lies in the hands of regional environmental officers confronted by an environmental and societal challenge.

Recommendations

Considering the national plan to control and manage lionfish invasion, potential management actions required for the Colombian Caribbean region may further focus on raising the community’s awareness by introducing the lionfish problem and the consequences of its invasion. Removals on a wider scale can be promoted by consumption of lionfish on a local and commercial scale. Public outreach should focus especially on lionfish as a good candidate for human nutrition. Morris et al. (2011) reported a relatively high content of lionfish fillet yield (30.5%) comparable to groupers, graysbys, and coneys. Lionfish meat had also higher content of essential n-3 fatty acids and a relatively low amount of saturated fatty acids as compared to other marine reef fish species (e.g., red snapper, dolphinfish, blue fin tuna, triggerfish, grouper and tilapia; Morris et al., 2011). The authors suggested that public outreach should especially focus on education about lionfish invasion, handling and cleaning of P. volitans in order to minimize risks for envenomation (Morris et al., 2011).

The establishment of marine reserves can effectively protect larger fishes (Halpern, 2003) such as groupers that could prey on lionfish as reported for the fishing-restricted Exuma Cays Land and Sea Park/Bahamas by Mumby, Harborne & Brumbaugh (2011). As long as it is not clear whether native predators are able to effectively prey on lionfish, further controlled fishing restrictions especially on native apex predator populations will become imperative for lionfish invasion control.

The invasion of P. volitans in the Western Atlantic and the Caribbean is considered as one of the top global threats to conservation of biodiversity (Sutherland et al., 2010). Local lionfish populations may disturb functioning of coral reefs through high consumption of small herbivorous fishes, including parrotfishes (Albins & Hixon, 2008; Morris & Akins, 2009), thus indirectly promote the outcompeting of corals by naturally uncontrolled growth of seaweeds (Mumby et al., 2006; Mumby & Steneck, 2008; Lesser & Slattery, 2011). Under the combined effects of overfishing, lionfish invasion (Albins & Hixon, 2011), global climate change (Hoegh-Guldberg, Ortiz & Dove, 2011), and local environmental degradations, the future of coral ecosystems is severely endangered (Jackson, 2010) in the Western Atlantic and Caribbean.

We acknowledge Juan F. Lazarus-Agudélo, Corvin Eidens, Christian M. Díaz-Sanchez, Johanna C. Vega-Sequeda, and particularly Julian Rau for SCUBA diving and assistance during the field trips. We thank the staff of Instituto de Investigaciones Marinas y Costeras ‘José Benito Vives de Andréis’ (Invemar) in Santa Marta, Colombia, especially Diana I. Gómez-López and Carolina Jaramillo-Carvajal for organizational support. We acknowledge the administration of the Tayrona National Natural Park for the kind collaboration.

Additional Information and Declarations

Competing Interests

Author Contributions

Field Study Permissions

Elisa Bayraktarov and Christian Wild are employees of the Leibniz Center for Tropical Marine Ecology; Javier Alarcón-Moscoso, Andrea Polanco F. are employees of the Instituto de Investigaciones Marinas y Costeras ‘José Benito Vives de Andréis’ (Invemar).

Elisa Bayraktarov conceived and designed the experiments, performed the experiments, analyzed the data, contributed reagents/materials/analysis tools, wrote the paper, prepared figures and/or tables, reviewed drafts of the paper.

Javier Alarcón-Moscoso performed the experiments, contributed reagents/materials/analysis tools, wrote the paper, reviewed drafts of the paper.

Andrea Polanco F. performed the experiments, wrote the paper, reviewed drafts of the paper.

Christian Wild conceived and designed the experiments, wrote the paper, reviewed drafts of the paper.

The following information was supplied relating to field study approvals (i.e., approving body and any reference numbers):

All necessary permits were obtained for the described study by Instituto de Investigaciones Marinas y Costeras ‘José Benito Vives de Andréis’ (Invemar) in Santa Marta, Colombia which complied with all relevant regulations (decree #302 and #309).

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
