# Peer review of "Spatial and temporal distribution of the invasive lionfish Pterois volitans in coral reefs of Tayrona National Natural Park, Colombian Caribbean"

_PeerJ, doi:10.7717/peerj.397_

## Round 0.1 · original submission · Minor Revisions

Overall, the referees are generally positive about the value and utility of the manuscript, but both make important suggestions for improvement. In particular, both reviewers suggest better incorporation of the available literature, and the second has extensive suggestions for improving the text throughout. Despite the length of the review, I feel that both reviewers make suggestions that are primarily editorial in nature, and so I expect that I will be able to evaluate your revised manuscript without requiring additional review. I want to be clear, however, that I agree with the comments of the referees and expect the manuscript to be revised appropriately before it will become acceptable. Also note that the second referee has also included an example figure which they believe will clarify your finding as a possible addition to Fig. 2. I expect that the suggested revisions should be relatively straightforward with a careful revision of your manuscript, and I look forward to seeing your revised submission.

Reviewer 1 ·

Basic reporting

No Comments

Experimental design

No Comments

Validity of the findings

I would like to see more discussion of the findings. The authors state that they saw no seasonal patterns, but there do appear to be large fluctuations in lionfish abundance, particularly in Guyraca Bay. Do they have any explanations for these? Or for why there are such large error bars on the Guyraca abundances? I would also like to see the size class abundances over time, particularly at Guyraca. Are the large fluctuations related to recruitment of juveniles? Influxes of adults? Finally, how does their pattern (or lack thereof) in abundance/size classes compare to seasonal studies in other related fishes? Or other invasive populations or native populations of lionfish? (See http://caribbeanfmc.com/pdfs/reni-Fisheries%20Independent%20-%20Final%20Report.pdf and http://re.indiaenvironmentportal.org.in/files/file/fishes%20in%20the%20Vellar%20estuary.pdf)

Additional comments

This is a good paper, and I believe is worthy of publication. However, it needs to be tied in better to our current knowledge of invasive populations, and could use further discussion.

Reviewer 2 ·

Basic reporting

see general comments

Experimental design

see general comments

Validity of the findings

see general comments

Additional comments

Bayraktarov et al. lionfish in Columbia
This is an interesting study to add to the growing literature documenting the lionfish invasion. Importantly, it illustrates monthly abundance data for a relatively recently invaded region (Columbia), includes accompanying size structure information (that should be better illustrated, see below), and also includes removals (whose methods need to be better described). With all the talk of removals being the best approach to lionfish management, this study also has the interesting aspect that despite monthly removals, these were not found to be effective in reducing lionfish abundance. Yes, lionfish are relatively easy to spear or remove by net but the generality of their habitat and prey utilization means that completely effective removals are virtually impossible and really a waste of resources without a sustained and repeated removal program. Even then, nothing will prevent lionfish from recolonizing an area (by either adult movement or larval delivery) so only very discrete areas that are fortunate to have a network of staff that can repeatedly return to continue removals will realize their benefits. Is this scenario possible and worthwhile? Can it be sustained? How long? Forever? Is this realistic and does it represent time and money well spent? Recent studies documenting success with removals of various levels on patch reefs cannot also apply to continuous reef habitat such as the fringing reefs that appear to be a feature of TNNP, and many other locations. As Frazer et al 2012 advocate, long term studies with adequate replication of removal and control sites from continuous as well as patch reef habitats are necessary to assess the ecological outcomes and feasibility of removals. The study should be published but requires revision which would improve its message.

Specific comments:
I’ve made a few grammatical comments because I think that English may not be the native language of the authors. These comments simply make the text read a little easier and are interspersed with the content comments.
Ln36-replace got with was
Ln36-replace ‘80s with 1980’s
Lns38-44: I don't think we can properly say that high fecundity is an advantage over native fauna. Many reef fish in native and invaded range display prolonged spawning with annual fecundity in the tens of millions; also, a planktonic life cycle and predatory habit are not advantages over native fauna as many native fish also share these characters; together, however, these traits make for increased probability of invasion success, but I don't think it's correct to speak to any of them as representing an advantage over native fauna. This would imply that native fauna have lower fecundity (not true), inferior dispersal mechanisms (not true), inefficient predatory habits (not true), etc. This section should be reworded.
Ln44-make lionfish succeed needs to be rewritten and supported with literature. The current explanation is too simplified
Ln50-reference fig 1 after TNNP
Ln52-where are these patches in relation to Chengue Bay, mentioned in next section?
Ln52-juvenile P. volitans, but Morris 2009 found 50% of male lionfish to be sexually mature at 10 cm TL
Ln54- can you give us some more information about this biodiversity? Is it elevated relative to Columbia, the Caribbean? be more specific; what is the management for the park? is fishing allowed? be more specific about levels of protection at the park
Ln58-insert (2012) after year
Ln64-comma after distribution
Ln65- maybe include a bit more detail about Kimball, which was lab-based; for example, Kimball et al didn't show an effect of temp on distribution of lionfish but as currently written that's the way the sentence reads; they predicted where lionfish might be distributed based on temp
Ln66-insert (removals) after mgmt. actions
Lns82-89: as currently written it is unclear that dry season is characterized by upwelling, high winds, high salinity, decreased temp; please substitute “characterized by” in place of “going along with a”; could also include during the dry season after deg C to make it clear for the reader
Ln87-delete “reports on the” and “characterize the oligotrophic…of upwelling”; insert after chl a “during periods of upwelling (dry) result in mesotrophic conditions, compared with oligotrophic conditions during the non-upwelling period (rainy).”
Ln90-substitute “can be found” for “are.” Also, more info about habitats, potential prey resources, fish communities within each bay would be nice, especially given different results of lionfish in different bays; why were CI and GA similar, and the other bays different? Are there data that can help you illustrate how these bays are similar or different with regard to these previous (or other) factors?
Ln92-delete ‘to coral reefs’
Ln98- I don’t think western and eastern should be capitalized
Ln99- what was the distance separating the 3 replicated transects from each other? Was 25 min spent on each transect or (as currently written) on each set of three replicates? If the latter, I would want to spend 25 min on each transect and believe that 25 min per 3 would not give enough time to accurately sample often cryptically located lionfish in a 250 m2 area
Ln105-insert TL after ‘body lengths’
Ln110- on line 75-76 the acronym for Invemar is defined differently
Ln111-substitute spearing and with nets… or spearing and netting…
Ln112- what was the maximum and minimum for these variable depths; we also need to know how much effort was spent each month on removals; what was the goal of removals, to remove all lionfish seen on a single dive, a single day, multiple days? did removals continue until no further lionfish could be seen? We need to have a better idea of effort. How many people participated in removals? How many people participated in regular surveys?
Ln113-unregistered removals? does this mean that untracked removals were taking place? If so, what is your sense of the effort of these removals? Do you have any general estimate for how many dive boats visit the area, fishermen, etc?
Ln123- substitute …resolution, all lionfish transect data were aggregated per month resulting...
Ln129 substitute …data with the software R…
Ln134-…before and after removal in the bays Chengue…
Ln135-delete sentence beginning with “The graphical…” to the end of the paragraph
Ln140- does this include the # of lionfish removed or is 123 the # of individuals seen; then how many were removed? Please clarify
Ln144-delete monthly
Lns144-152: I'd like the average before and average after removal data in a fig; maybe a grouped bar chart with a bar for each bay, with removal bays indicated and each bay grouped, before and after, total of 8 bars; mean density per ha and SE on Y; see attached fig for example; such a fig would help illustrate to readers, for example, that post removal, one removal bay CI ns different from non removal GA; removal CI sig greater than non removal NE; removals sig diff from each other;
Ln145-delete ‘for the registered individuals’
Ln155-insert ‘(a removal bay)’ after Chengue
Ln154 paragraph-How about a topic sentence such as ‘Our monthly lionfish censuses demonstrated temporal and spatial variability in lionfish abundances among TNNP bays, which varied between 0 and 16.7 ind per ha.’
Lns164-167: these two statements seem to disagree; first says that 40 cm fish were found in Cinto and Neguanje ; then second sentence says Cinto and Gayraca had 20-25 cm as their largest size class and N had 15-20 cm; please clarify; also, I think an informative figure, maybe a second panel of fig 2 would be same x axis but with y showing mean +-SE lionfish size in each bay for each month; that would give the reader an idea if lionfish were moving or recruiting into various bays throughout the year
Ln167-delete registered; substitute ‘observed’ for ‘individuals’
Ln168-substitute ‘17.5 cm TL’ for ‘17’ and ‘50% maturity for females’ for ‘mature females’
Lns168-170: how did this estimate of mature fish change after removals?
Ln185- substitute ‘removal’ for ‘controlled’; Many people think of control as an experimental treatment where a manipulation does not occur, or does occur but to control for the manipulation effect. Since CH and CI were bays where treatments/conditions were manipulated, how about referring to manipulated bays as removal bays and bays were removals did not take place as control bays?
Ln187-delete ‘between the abundance data’
Ln188-substitute ‘…months of monitoring in the four bays (total of n=282 transect replicates) were pooled to calculate an annual mean of 2.9 ± 0.9 ind ha-1 (± SE) for the TNNP region.’
Ln195-spell out acronym again for TNNP here at start of discussion
Ln198-do you mean sexually reproduce? Not sure what recruit sexually means. If so, substitute ‘…able to sexually reproduce, based on Morris (2009) who reported 17.5 cm TL as the size of 50% maturity for females.’
Ln203-not sure the table is necessary, especially where different methods are involved
Ln207-substitute ‘of’ for ‘off’
Lns210-211: unlimited is too strong an adjective. If you're going to make that statement you need to support it with references that show that growth and reproduction are unlimited. I don't think such studies exist
Ln214-As written, I read this to mean that the researchers watched the actual acts of predation, which is not the case. Maybe rephrase, for example: ‘...were captured with lionfish in their stomachs’, or something similar that indicates lionfish were observed in the stomach contents
Lns220-221: substitute ‘abundance’ for success and delete population
Lns222-223: ‘stress the necessity immediate management actions…’; but your mgmt actions showed no effect so why would you advocate immediate mgmt actions? Also, you should review and also cite the following to indicate that the idea of biocontrol is the subject of an active debate: Bruno JF, Valdivia A, Hackerott S, Cox CE, Green S et al. (2013) Testing the grouper biocontrol hypothesis: A response to Mumby et al. 2013. PeerJ PrePrints 1:e139v1 http://dx.doi.org/10.7287/peerj.preprints.139v1,
as well as Mumby PJ, Brumbaugh DR, Harborne AR, Roff G. (2013) On the relationship between native grouper and invasive lionfish in the Caribbean. PeerJ PrePrints 1:e45v1 http://dx.doi.org/10.7287/peerj.preprints.45v1
also see Stephanie J. Green, Nicholas K. Dulvy, Annabelle L. M. Brooks, John L. Akins, Andrew B. Cooper, Skylar Miller, and Isabelle M. Côté In press. Linking removal targets to the ecological effects of invaders: a predictive model and field test. Ecological Applications. http://dx.doi.org/10.1890/13-0979.1
ln225- substitute ‘characterized by’ for ‘going along with’
ln227-substitute ‘did not appear to affect’ for did not affect
ln228-substitute ‘the abundance of lionfish at TNNP.’ for ‘the established lionfish population’
ln228- were these temperatures actually measured in this study or is it the typical decrease in temp associated with upwelling? Also, maybe temp had an effect but you didn't measure it or couldn't detect it? Stating that you didn't detect an effect of temp is different than saying there was no effect; and the fact that kimball's work was lab based means those results may be very different than what you might observe in the field
ln230-substitute ‘…reported in the laboratory to…’ for ‘reported to’
ln238-again, a fig illustrating lengths would be helpful to your point being made here
ln240-substitute ‘thereby may indicate’ for ‘thereby indicates’; another potentially informative fig would be a 4 panel fig illustrating a histogram of size classes censused and removed from each bay; this would help illustrate whether certain bays harbor different sizes of lionfish relative to others
ln242-‘…targeting predominantly larger adult…’; this needs to be put in methods in removal section; I would say may indicate habitat preference, but as you say, this is also where you removed lionfish; so maybe a histogram of lionfish sizes prior to removal for all bays would give you an indication of whether juveniles preferentially settle to Ch. Also, do the prevailing currents supply Ch with juveniles before the other bays, since Ch is the easternmost bay? Are oceanography and prevailing currents somehow related to the distribution of juveniles?
Lns250-252: yes, but can previous studies at TNNP tell us anything about these factors?
Ln254-substitute ‘removal’ for ‘control’
Lns256-257: ‘…to preserve the condition of Caribbean coral reef ecosystems…’; is this possible? Does TNNP have the resources to do this?
Lns262-263: exactly. Who is going to remove lionfish outside park boundaries? The park may spend great resources in removing lionfish from within its boundaries and just beyond, thriving populations of lionfish will be available to recolonize; lionfish do and can move (Jud and Layman's 2012 study showing site fidelity used juveniles while Green et als 2011 showed a minimum movement of 130 m), and currents can repeatedly deliver larvae. Lionfish will be a never ending problem
Lns269-270: yes, this is a good idea
Ln270-substitute ‘…commercial scale.’ and delete from ‘as suggested…’ through ‘The authors suggest that’; Then start new sentence: ‘Public outreach should especially focus…’
Ln281-‘will become imperative for lionfish control’; but we don't know whether or not native predators effectively prey on lionfish yet; experimental field and laboratory studies examining native predator-lionfish interactions are needed before we state that fishing restrictions are imperative
Lns286-287: also see Lesser and Slattery 2011
Lns287-291: This last sentence is too vague and broad, and repeats some of previous sentence
Fig 2: represent the removal bays as either both solid or both open so reader can easily compare with control (non-removal bays)
Fig 3: insert ‘aggregate of 4 bays’ after SE in caption
Table 1: maybe include a column with first record of invasion; might also want to remove studies that used different methods from the present;

Annotated reviews are not available for download in order to protect the identity of reviewers who chose to remain anonymous.

---

## Round 0.2 · accepted · Accept

Thanks for your thorough revision of the manuscript and attention to the referee feedback. You have addressed each of their comments to my satisfaction and I am happy to move your manuscript forward at this point. There are some remaining grammatical issues that should be possible to deal with but do not require additional review.